# Missing Link Between Molecular Aspects of Ventricular Arrhythmias and QRS Complex Morphology in Left Ventricular Hypertrophy

**DOI:** 10.3390/ijms21010048

**Published:** 2019-12-19

**Authors:** Ljuba Bacharova

**Affiliations:** 1International Laser Center, 841 04 Bratislava, Slovakia; ljuba.bacharova@ilc.sk; Tel.: +421-905-401-634; 2Institute of Pathophysiology, Medical School, Comenius University, 841 04 Bratislava, Slovakia

**Keywords:** left ventricular hypertrophy, ventricular arrhythmia, electrical remodeling, non-specific ECG predictors, QRS complex morphology

## Abstract

The aim of this opinion paper is to point out the knowledge gap between evidence on the molecular level and clinical diagnostic possibilities in left ventricular hypertrophy (LVH) regarding the prediction of ventricular arrhythmias and monitoring the effect of therapy. LVH is defined as an increase in left ventricular size and is associated with increased occurrence of ventricular arrhythmia. Hypertrophic rebuilding of myocardium comprises interrelated processes on molecular, subcellular, cellular, tissue, and organ levels affecting electrogenesis, creating a substrate for triggering and maintaining arrhythmias. The knowledge of these processes serves as a basis for developing targeted therapy to prevent and treat arrhythmias. In the clinical practice, the method for recording electrical phenomena of the heart is electrocardiography. The recognized clinical electrocardiogram (ECG) predictors of ventricular arrhythmias are related to alterations in electrical impulse propagation, such as QRS complex duration, QT interval, early repolarization, late potentials, and fragmented QRS, and they are not specific for LVH. However, the simulation studies have shown that the QRS complex patterns documented in patients with LVH are also conditioned remarkably by the alterations in impulse propagation. These QRS complex patterns in LVH could be potentially recognized for predicting ventricular arrhythmia and for monitoring the effect of therapy.

## 1. Introduction

Ventricular arrhythmias, especially ventricular tachycardia and ventricular fibrillation, are life-threatening arrhythmias, frequently leading to sudden cardiac deaths. Therefore, identification of the arrhythmia predictors is of the utmost clinical importance.

There is rapid progress in the field of arrhythmology. It is the most growing part of electrocardiology, both in the basic science as well as in clinical electrocardiography and electrophysiology. There is an increasing body of knowledge and details on the substrate triggering and maintaining arrhythmias, such as changes in ion channels, action potentials, gap junctions, connexin, fibrosis, etc., revealing details on molecular mechanisms of ventricular arrhythmias. This knowledge of structural, functional, and electrical alterations serves as a background for identifying patients at risk of ventricular arrhythmias and creates a basis for development of risk stratification scores [1,2,3,4,5].

Ventricular arrhythmias, however, are manifestations of electrical instability. The unique method of recording electrical activity of the heart is electrocardiogram (ECG); in clinical practice, the standard 12-lead ECG is generally used.

There is a considerable gap between the detailed knowledge on the molecular level and the clinical possibilities and interpretation of ECG findings in evaluating the effect of antiarrhythmic therapy. Reducing this gap is vital since the ultimate evaluation of any targeted antiarrhythmic therapy is based on the results of clinical studies.

Currently, there are several ECG findings that are recognized as ventricular arrhythmia predictors, and their interpretation is related to altered electrophysiological processes and to our understanding of ventricular arrhythmias mechanism. They are not specific for a certain clinical diagnosis and include, e.g., prolonged QRS complex duration, prolonged QT interval/corrected QT interval (QTc), early repolarization, and the fragmented QRS complex (fQRS). 

In contrast, the progress and detailed knowledge in terms of electrogenesis are not reflected in the interpretation of P-QRS-T waveforms analysis, where the traditional interpretation still persists. The aim of this opinion paper is to discuss possible links between QRS complex morphology and ventricular arrhythmias based on molecular and electrophysiological processes and to discuss ECG patterns that could potentially indicate abnormal electrical substrate pertinent to conditions necessary for re-entry to occur. The links between ECG diagnostic criteria based on underlying processes affecting electrogenesis are demonstrated using an example of left ventricular hypertrophy (LVH).

## 2. Left Ventricular Hypertrophy

LVH is defined as an increase in the left ventricular size/left ventricular mass (LVM), and therefore the dominant methods for detecting the left ventricular (LV) anatomical enlargement are imaging methods. 

Numerous studies have documented the association of anatomical left ventricular hypertrophy (LVH) with ventricular arrhythmias [6]. It was shown in the Framingham study [7] that the presence of echocardiographic LVH in the general screening population was associated with increased risk of ventricular arrhythmia. Hypertensive patients with LVH have higher risk of developing ventricular arrhythmias [8], and even in patients with mild to moderate arterial hypertension, the echocardiographic LVH, particularly eccentric LVH, is associated with higher risk of ventricular arrhythmia [9]. Higher risk of developing ventricular arrhythmias is documented in patients with valvular heart disease [10,11], which is reduced after surgical treatment [12].

In clinical practice, the term *remodeling* in relation to LVH is understood mainly on the anatomic scale and recognizes concentric remodeling, concentric hypertrophy, and eccentric hypertrophy. However, it is obvious that it is not the simple increase in LV size and/or LVM that creates conditions for arrhythmias. Triggering and maintaining arrhythmias require electrophysiological conditions, i.e., arrhythmogenic substrate that is generally referred to by the term electrical remodeling.

The hypertrophic rebuilding of myocardium is associated with a complex interrelated structural and functional rebuilding of ventricular myocardium, including complex changes in active and passive electrical properties. It is not a uniform process, but it shows considerable heterogeneity, including the size of hypertrophic cardiomyocytes [13,14], membrane currents [15], the connexin expression, as well in action potential characteristics [16,17].

The basic characteristic in LVH on the microscopic scale is the increase in size of individual cardiomyocytes documented in numerous animal and human studies. Additionally, the hypertrophied cardiomyocytes differ from the normal cardiomyocytes not only in diameter and length but also in branching and number of connected cardiomyocytes [18,19,20,21]. These structural changes affect upstroke velocity of the action potential, the action potential propagation, conduction velocity, and cell-to-cell impulse propagation [22,23,24].

An integral part of the hypertrophic rebuilding of myocardium is the considerable changes in interstitium. The main processes include fibrosis, inflammation, degenerative changes, and apoptosis [25]. Fibrosis is a diffuse process and can also create fibrous tissue depositions, creating localized areas, e.g., the mid-wall fibrosis described in patients with LVH [26,27]. Additionally, inflammation also creates localized areas of edema and accumulated blood cells [28,29]. Diffuse and localized areas of pathologically changed intercellular matrix contribute thus to the heterogeneity of the electrical impulse propagation. These changes represent potentially arrhythmogenic substrate and correlate with ventricular arrhythmias and sudden cardiac death [30,31].

At the molecular level, an extensive body of evidence has been published. There are a variety of alterations in ion channels, gap junctions, connexin43, i.e., the dominant molecular processes affecting electrogenesis. For the relation of these processes to the QRS complex see e.g., [32].

Animal as well as human studies show a variety of results on the main depolarizing currents affecting the resting potential and the action potential configuration: the fast sodium current ***I**_Na_*, L-type calcium current ***I**_Ca(L)_*, and Na^+^-Ca^2+^ exchanger ***I**_Na/Ca_*. A decrease in ***I**_Na_* density has been documented [33,34], but on the other hand, its increase has also been described [35]. Similarly, a decrease in ***I**_Ca(L)_* current has been shown [36,37], but an increase [38,39], or no difference has also been reported [40,41]. More consistent are findings regarding ***I**_Na/Ca_*, where an increase in ***I***_(Na/Ca)_ density has been reported [33,42]. The ion channel remodeling is reviewed in detail by e.g., [43,44,45,46].

Alterations in ion channels are reflected in changes in resting membrane potential, as well as in action potential (AP). The results related to the depolarization phase of AP are not consistent. On one hand, no differences in resting membrane potential, the action potential upstroke velocity, or amplitude are documented [47,48]; on the other hand, significant changes are reported [39,49,50]. Also, there is not a consistent pattern in conduction velocity changes of hypertrophied cardiomyocytes [51,52]. The variety of results can be due to differences in experimental models of LVH, species and preparation employed, the severity of hypertrophy, methodological aspects, etc.; however, they create a solid base of evidence on the changes influencing electrical characteristics of hypertrophied cardiomyocytes and myocardium.

Gap junctions, the low-resistance intercellular structures, enable the functional connection between cardiomyocytes, and they are crucial for uninterrupted propagation of electrical impulse. The density and distribution of the gap junctions (GJ) in LVH are significantly changed, as has been documented in human studies and animal studies, including cell cultures and genetically modified models of LVH (Figure 1 and Figure 2). The intercellular coupling is diminished and disorganized, creating a basis for altered ventricular activation, i.e., the structural and functional substrate for developing lethal arrhythmias [53,54,55,56]. 

Connexin43 is the dominant isoform in the adult ventricles. Its expression and distribution in LVH is affected already in the early stage of LVH development, as is documented in animal and human studies, as well as in cultured cardiomyocytes [53,54,56,57,58,59]. Additionally, the gap junction conductance can be affected by additional factors such as pH and Ca^2+^ ion concentration [60].

Structural and functional alterations at the molecular level lead to changes in active and passive electrical properties of myocardium. Their interrelation is presented in Figure 3. Regarding the action potential (AP) of the hypertrophied cardiomyocyte, changes in AP shape and duration depend on the functional expression of depolarizing and repolarizing currents [43,44]. The documented changes are mostly related to the repolarization phase, especially its prolongation [61,62]. However, there are significant changes also in the depolarization currents [32]. Additionally, it was shown that the process of uncoupling may also induce changes in AP duration [63].

## 3. Non-Specific ECG Predictors of Ventricular Arrhythmia in LVH

### 3.1. QRS Complex Duration

The prolonged QRS complex duration is a significant predictor of ventricular arrhythmia [64,65,66]. The interpretation of the relation between the prolonged QRS complex duration and occurrence of ventricular is quite straightforward and logical: the slowed activation of the ventricles (ventricular depolarization) prolongs the QRS complex duration, and pathophysiologically, the altered conduction of the ventricular myocardium is one of the characteristics of the arrhythmogenic substrate. In left ventricular hypertrophy, the QRS duration is prolonged and is associated with ventricular arrhythmia and sudden cardiac death [28,67,68]. Two main factors affect the QRS duration: anatomical—the increased left ventricular mass, i.e., a longer trajectory for the electrical impulse to pass; and functional—the slowed velocity of impulse propagation. The prolongation of the QRS complex, as well as the delayed intrinsicoid deflection via the leads V5/V6, is incorporated into the Romhilt-Estes score for LVH detection [69]. 

However, in cases when the increased QRS complex duration is associated with certain typical QRS morphology, the interpretation of these findings is shifted from the altered conduction of the ventricular myocardium to the conduction disorders of the conduction system—namely, to the ECG diagnosis of left bundle branch blocks (LBBB). These QRS patterns are also associated with a higher incidence of cardiac mortality in patients with LVH [70]. 

Using the term *LBBB*, the understanding of this typical QRS pattern is implicitly reduced to the block in the conduction system, and the alteration of the ventricular working myocardium is not considered. This modified assumption then contrasts with the finding of increased presence of the conduction disorders with more severe ventricular hypertrophy or in myocardial infarction [71,72]. Using computer modeling, it was shown that the typical LBBB pattern can be a manifestation of slowed conduction velocity in the working myocardium, even when there is no block in the conduction system [73,74]. This is consistent with clinical findings of the increased association between LBBB patterns and clinical condition with severe alteration of working myocardium, such as severe LVH or myocardial infarction [75,76,77].

On the other hand, however, there are also conflicting results showing that the relation between QRS complex duration, conduction disorders, and ventricular arrhythmias is not that straightforward [78,79]. 

### 3.2. QT Interval

It has been shown that QT interval prolongation above the upper normal limits and variability of QT interval are associated with an increased risk of ventricular arrhythmia [80]. Both increased QT interval and QT dispersion have been found in LVH patients with different anatomical types [81,82] and are associated with the occurrence of ventricular arrhythmias and sudden cardiac deaths [83,84].

QT interval and QT dispersion are usually reported as parameters of repolarization. However, the QT intervals represent the time of both depolarization and repolarization. In clinical studies, less (or even no) attention is devoted to the contribution of the depolarization into the QT interval parameters. 

In our simulation study [85], we presented the QT/QRS ratio with the aim to differentiate the QT prolongation due to primary depolarization prolongation (if the values of the ratio decreases) and repolarization prolongation (if the values of the ratio increases). The QT/QRS ratio was used in an experimental study [86] and was shown as a useful parameter in predicting potential risk of drug-induced ventricular arrhythmia. Later, Robyns et al. [87] showed its potential diagnostic usefulness in differentiating the predisposition to torsade de pointes (TdP) and non-TdP. The increase in the QT/QRS ratio was associated with the predisposition to TdP, and its decrease was associated with the predisposition for non-TdP-mediated VT/VF. Those results were not discussed in terms of pathophysiology, so considering a possible pathophysiological background, in the case of the predisposition to TdP, the increased QT/QRS ratio could point to the prolonged repolarization; and in the case of non-TdP-mediated VT/VF, the decrease in the QT/QRS ratio could point to the prolonged depolarization.

Technically, the QT interval is measured from the onset of the QRS complex to the end of the T wave, which could create a problem in cases with low amplitude T wave signal, both when manually as well as automatically measured. Very probably, this also contributes to the variety of recommended normal values.

Summarizing, it has been shown that QT interval as a predictor of ventricular arrhythmias has its potential; however, it has also its challenges in interpretation as well as in its precise measurement. [65,88].

### 3.3. Early Repolarization

Early repolarization (ER) refers to changes at the junction of the offset of the QRS complex and beginning of the ST segment: the J-waves or J-point elevation. It is frequently seen in young athletes, and it was shown to correlate with concentric remodeling. Although it is considered benign, it is assumed that it is related to structural and electric modifications induced by training [89]. Under pathological conditions, ER is associated with ventricular arrhythmia and sudden cardiac death [90,91,92,93,94,95]. It is assumed that the pathophysiological mechanism triggering ventricular arrhythmias is the regional heterogeneity of cardiomyocytes refractoriness caused by alteration of repolarizing currents, including local differences in I_to_ current density [96,97,98]. However, controversial results are also reported [99].

The definitions of the ECG findings in ER vary; additionally, the ER abnormalities are identified at the junction of the QRS complex and the ST segment deviation. Therefore, there are discussions about whether these abnormalities represent ER, late depolarization, or other factors [100,101].

### 3.4. Late Ventricular Potentials/Signal Averaged ECG

The term *ventricular late potentials* (LP) refers to low-amplitude, short-duration signals that occur after the QRS complex. For their detection, a special electrocardiographic technic, signal-averaged electrocardiography (SAECG), is used, which allows reducing interference and revealing low-amplitude signals at the end of the QRS complex.

The presence of LP is associated with scars and structural abnormalities of myocardium, i.e., structural conditions for re-entry, and they may indicate a predisposition to ventricular arrhythmias and sudden cardiac death [102,103,104,105]. LPs were frequently observed also in patients with LVH [106,107,108]. However, their predictive value is also an open question [9,109].

### 3.5. Fragmented QRS Complex

Fractionation or fragmentation of the QRS complex (fragmented QRS complex—fQRS) is defined as the presence of R’ wave or notching of R or S wave in the presence of narrow QRS. It indicates heterogeneous depolarization of the ventricular myocardium that can occur due to ischemia, fibrosis, scar, or coronary microvascular dysfunction. Fragmented QRS complex is a significant and independent predictor of cardiovascular arrhythmia [110,111,112,113]. Fragmented QRS complex is frequently seen in patients with LVH, and it was reported to be associated with ventricular arrhythmia [114,115,116].

### 3.6. Body Surface Potential Mapping

Body surface potential mapping (BSPM) provides total information on the temporospatial distribution of electrical potentials on the surface of the chest. As compared to the limited information provided by the standard 12-lead ECG, it enables detection of local electrical events or local conduction disturbances, reflecting heterogeneity of underlying ventricular activation and repolarization [117]. Thus, it can identify the signs of susceptibility to arrhythmias and the site of origin of the arrhythmia. An additional advantage is the possibility to derive epicardial maps noninvasively (ECG imaging) from the distribution of surface potentials [118].

Although BSPM is a promising method, its utilization in clinical practice is still limited because of its demanding recording and interpretation, as well as of the need for standardization of lead systems and methods of BSPM data analysis.

Taken together, the interpretation of the above-mentioned ECG predictors of VA is related to altered electrophysiological processes and to our understanding of ventricular arrhythmia mechanisms. They are not specific for left ventricular hypertrophy, but they refer to underlying processes that can be present in a variety of cardiac pathology. In spite of their limitations, they represent challenging parameters of electrical remodeling. As was reviewed, LVH is associated with higher prevalence of these non-specific predictors of ventricular arrhythmia resulting from significant rebuilding of myocardium-affecting electrogenesis that is potentially arrhythmogenic.

## 4. Waveform Interpretation in LVH

Paradoxically, there is a substantial gap between this detailed knowledge on molecular, subcellular, cellular and tissue levels on one hand, and clinical interpretation of ECG findings in LVH patients on the other. Consequently, the detailed knowledge on the electrical properties of hypertrophied myocardium contrasts with the clinical ECG diagnosis of LVH. The ECG diagnosis of LVH is mostly reduced on the increase in the QRS complex amplitude, so-called voltage criteria [119]. The anatomical aspect of LVH is stressed, and it is postulated that the bigger (hypertrophied) left ventricle generates a stronger cardioelectric field, which is reflected in the increase in the QRS complex amplitude. The possible effect of the electrical remodeling on the QRS complex is not considered.

### 4.1. ECG Criteria on LVH

Since LVH is defined as an increase in LV size/mass, the gold standard for comparative studies on LVH is left ventricular size/mass. The role of ECG in LVH diagnosis (ECG-LVH) is thus seen also in estimating the LVM. If there is an agreement between the increased LVM and the increased QRS complex amplitude, these results are assigned as true positive (which represents the minority of LVH cases); if there is a disagreement, these results are earmarked as false negative. The meta-analysis by Pewsner showed a wide range of sensitivity and specificity of ECG-LVH criteria [120], documenting the weakness of this approach.

The increased QRS complex amplitude is indeed associated with the increased risk of ventricular arrhythmia [7]. However, since the prevalence of ECG-LVH is lower as compared to the LVH detected by e.g., echocardiography, the calculated risk is also lower, which led to a questionable conclusion that ECG-LVH is less sensitive for ventricular arrhythmias. On the other hand, however, it was shown that even the so-called false negative ECG results have the prognostic value in CV risk assessment [121]; therefore, the focus only on the increased QRS voltage is questionable.

The Romhilt–Estes score (R–E) is a more complex ECG-LVH criterion and considers also other ECG abnormalities additional to the increased voltage [70]. It was shown that the individual components of R–E scores differ in predicting different CV outcomes and have independent predictive ability. It can be assumed that they reflect different pathophysiological aspects of the complex hypertrophic rebuilding [122].

It is obvious that if ECG records the electric field of the heart, the electrical properties of myocardium must be considered, especially in the case of ventricular arrhythmias, and the role of ECG-LVH diagnosis cannot be reduced and simplified to the estimation of the increase in LVM.

### 4.2. ECG Findings in LVH Patients

The increased QRS voltage is only one of the whole spectrum of QRS patterns observed in patients with LVH. It occurs in a minority of LVH patients, and its sensitivity ranges from 0% to 60% [120].

Additional to the increased QRS amplitude, there are other deviations from the normal QRS patterns in patients with LVH, such as the leftward shift of the electrical axis in the frontal plane, prolonged QRS duration, as well as ST segment and T wave changes. The intraventricular conduction defects, such as left bundle branch block (LBBB), left anterior fascicular block (LAFB), and especially the incomplete LBBB, are frequently observed. Finally, there are those false negative results, i.e., apparently normal QRS complexes (Figure 4).

Considering the multiple and interrelated changes affecting the electrical properties of hypertrophied myocardium, we used computer modeling to study their effect on the resultant QRS pattern [73]. The effect of slowed conduction velocity on the QTS pattern in three anatomical types of LVH was studied. As a composite parameter of electrical properties, we used slowed conduction velocity corresponding to published data [123,124,125]. 

It was shown that the increased LVM and the anatomical type of LVH are not the main factors affecting the QRS complex and that the increase in LVM does not result in proportional increase in QRS amplitude. It is the diffuse slowing of conduction velocity in combination with the anatomical types of LVH that affect QRS amplitude, duration, and morphology, and we observed the whole variety of QRS patterns that are seen in LVH patients. Interesting was also the effect of anatomical changes and/or conduction velocity slowing on the clinical ECG-LVH criteria that were affected differently. We also showed that QRS patterns consistent with QRS complex changes observed in patients with LVH can be seen also in a normal-sized heart if the slowed conduction velocity is slowed.

A frequent finding in different types of left ventricular is midwall fibrosis [26,27,126]. We showed in a simulation study that the midwall slowing in conduction velocity also modified the QRS complex morphology and resulted in QRS patterns usually interpreted as the effect of left ventricular hypertrophy, i.e., the increase in LV mass [127].

Summarizing, on one hand, there are documented details on molecular, structural, and functional changes in LVH; on the other hand, there is a simplified mechanistic approach used for diagnosing and evaluating/monitoring the effect of therapy. When talking about remodeling LVH in clinical practice, the anatomical types of LVH are considered: concentric remodeling, concentric hypertrophy, and eccentric hypertrophy. Although ECG is the unique method recording the electrical phenomena, in the case of LVH, its role is shifted to the estimation of anatomical aspects of LVH.

The possibilities for evaluating the effect of therapy in LVH in clinical practice are limited to imaging methods and ECG (if not talking about clinical outcomes). What is the desired effect of the therapy in LVH in relation to the ventricular arrhythmias? The reduction of the LV mass? The reduction of the QRS voltage? How are they related to expected effects of targeted therapy that are based on the detailed knowledge of the molecular processes? How is this detailed knowledge translated into the ECG diagnosis of LVH? To detect their early manifestation in electrocardiogram prior to the manifestation of ventricular arrhythmia? If left ventricular hypertrophy is the therapeutic target, then how does one best combine the arrhythmia biomarkers for monitoring the effect of therapy [128]?

## 5. Conclusions

There is a remarkable gap in knowledge between molecular details on one hand and diagnostic possibilities and ECG interpretation in clinical practice in the case of left ventricular hypertrophy on the other. It is documented that in LVH, the myocardium is considerably changed, and these changes are potentially arrhythmogenic. To decrease this gap is a real challenge for electrocardiography as well as for basic scientists. Electrocardiology needs to go beyond traditional diagnostic categories and to reconsider the classical interpretation of QRS-T patterns. Electrocardiology needs to focus primarily on the function of the heart as a source of cardio-electric field, considering and incorporating the current knowledge in electrophysiology and electrical remodeling under pathological conditions. There are no such results as false of confounding; it is the false expectation and lack of understanding of the processes, and consequently, the meaningful information is so far neglected. Basic scientists need to understand clinical challenges and help translate the extensive knowledge on molecular mechanisms into relevant clinical categories.

## Figures and Tables

**Figure 1 ijms-21-00048-f001:**
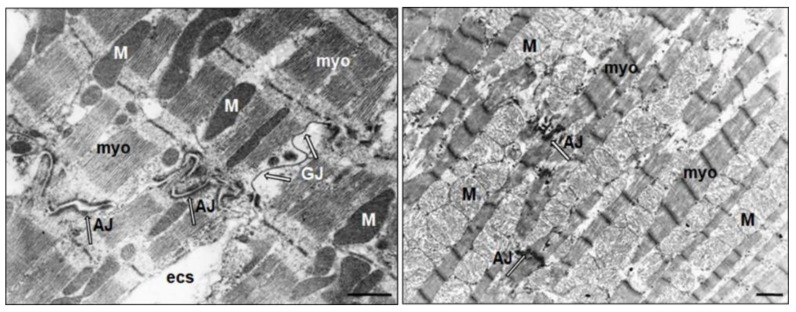
Representative electron microscopic images of cardiomyocytes from the hypertrophied left ventricle of spontaneously hypertensive rats. Left panel: hypertrophied cardiomyocyte exhibiting electron-dense mitochondria (M) and well-preserved intercalated disc with adhesive junctions (AJ: fascia adherens and desmoses), as well as gap junctions (GJ), as indicated by arrows. Right panel: hypertrophied cardiomyocyte exhibiting severe subcellular injury including electro-lucent mitochondria, lysis of myofibrils, and rudimentary intercalated disc with remnant adhesive junction and missing gap junctions. Bar = 1 micrometer. myo: myofibril; esc: extracellular space.

**Figure 2 ijms-21-00048-f002:**
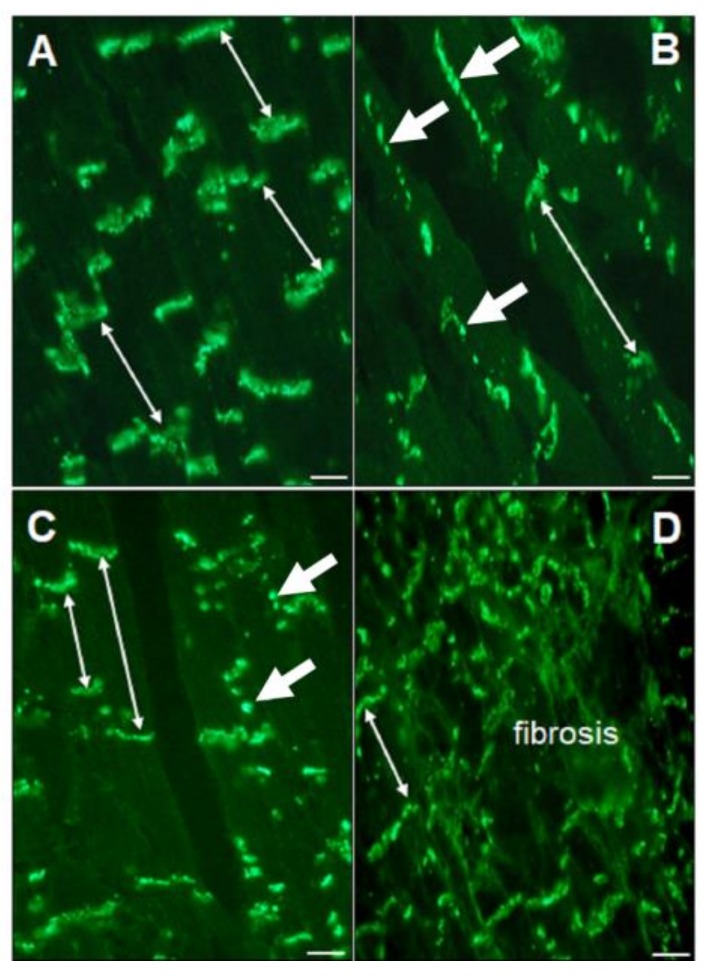
Variations in gap junctions in hypertrophied left ventricle documented in spontaneously hypertensive rats, as identified by immunodetection of Cx43. **A**: Dominant Cx43 distribution in the intercalated discs (thin arrows); **B**: Dominant Cx43 distribution in lateral sites of the cardiomyocytes (thick arrows); **C**: Both lateral and intercalated disc distribution is seen (thick and thin arrows); **D**: Marked disordered Cx43 distribution in the fibrotic area. Thin arrows also indicate longitudinal orientation of the cardiomyocytes. Bar = 10 micrometers.

**Figure 3 ijms-21-00048-f003:**
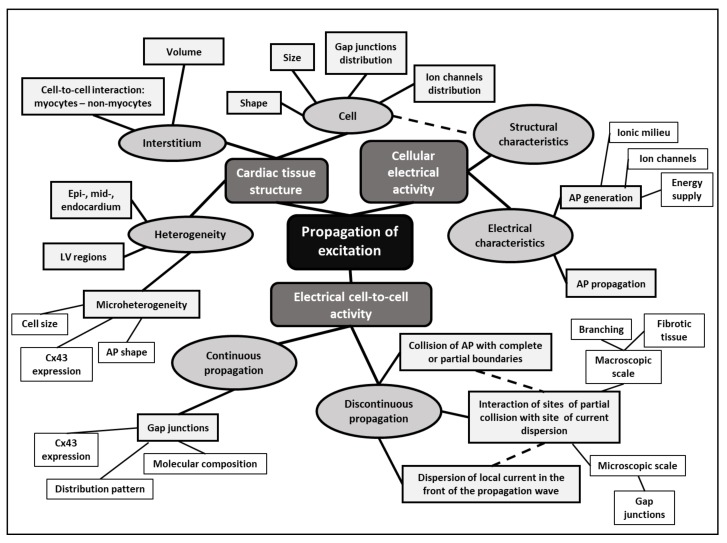
Mind map: factors involved in the impulse propagation in hypertrophied myocardium. Reprinted from Bacharova et al. 2007 [33] with permission from Wiley (modified). AP: action potential; Cx43: connexin 43.

**Figure 4 ijms-21-00048-f004:**
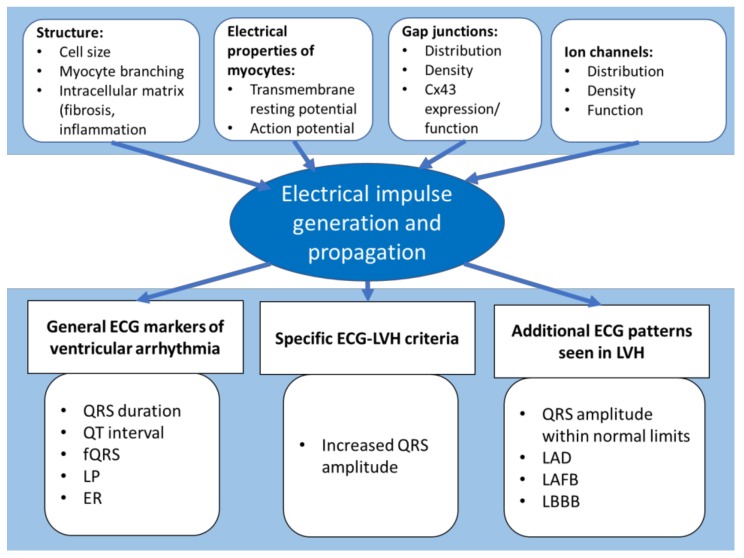
The main factors influencing electrogenesis in left ventricular hypertrophy and the variety of ECG manifestations related to ventricular depolarization (QRS complex). fQRS: fragmented QRS complex; LP: late ventricular potential; ER: early repolarization; LAD: left axis deviation; LAFB: left anterior fascicular block; LBBB: left bundle branch block.

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
