# Peer review of "Missing Link Between Molecular Aspects of Ventricular Arrhythmias and QRS Complex Morphology in Left Ventricular Hypertrophy"

_ijms, 2019, doi:10.3390/ijms21010048_

Round 1

Reviewer 1 Report

The authors presented the pathophysiological understanding of LVH from the perspective of structural and electrical remodeling, and suggested the usefulness and problems of diagnostics using ECG in clinical practice. They are the useful knowledge regarding the development of new therapeutic strategies and the effective use of therapeutic agents. However, concerning QRS and conduction velocities of LVH, the authors should describe the detailed information not only about gap junctions and connexin 43 but also about ion channels such as Na+ channels.

Minor comment

Figure 2: I couldn't distinguish between thick and thin arrows. Please make the difference between the two easy to understand.

Author Response

The authors presented the pathophysiological understanding of LVH from the perspective of structural and electrical remodeling, and suggested the usefulness and problems of diagnostics using ECG in clinical practice. They are the useful knowledge regarding the development of new therapeutic strategies and the effective use of therapeutic agents. However, concerning QRS and conduction velocities of LVH, the authors should describe the detailed information not only about gap junctions and connexin 43 but also about ion channels such as Na+ channels.

Thank you for suggesting me to include more detailed information on ion channels. When writing the manuscript I had a dilemma how much details to present on the ion channels remodeling in LVH and I was afraid that going into details the review would be not balanced. On the other hand I understand that it is the key topic. The knowledge on ion channels is really extensive and I did my best to compress it into a limited space to balance it with the other parts of the manuscript. I am sure other experts will provide more qualified details in this issue.

The following two paragraphs were added with corresponding references [34-53]:

Page 3, lines 100-115:

Animal as well as human studies show a variety of results on the main depolarizing currents affecting the resting potential and the action potential configuration : the fast sodium current INa , L-type calcium current ICa(L) and Na+-Ca2+ exchanger INa/Ca. A decrease in INa density was documented [34, 35], on the other hand its increase was described [36]. Similarly, a decrease in ICa(L current was shown [37, 38], an increase [39, 40], or no difference was reported [41, 42]. More consistent are findings regarding INa/Ca, where an increase in I(Na/Ca) density was reported [34, 43]. The ion channel remodeling are reviewed in details by e.g. [44-47].

Alterations in ion channels are reflected in changes in resting membrane potential, as well as in action potential (AP). The results related to the depolarization phase of AP are not consistent. On one hand, no differences in resting membrane potential, the action potential upstroke velocity and amplitude are documented [48, 49], on the other hand, significant changes are reported [40, 50, 51]. Also, there is not a consistent pattern in conduction velocity changes of hypertrophied cardiomyocytes [52, 53]. The variety of results can be due to differences in experimental models of LVH, species and preparation employed, the severity of hypertrophy, methodological aspects, etc., however, they create a solid base of evidence on the changes influencing electrical characteristics of hypertrophied cardiomyocytes and myocardium.

Chorvatova, A.; Hart, G.; Hussain, M. Na+/Ca2+ exchange current (I(Na/Ca)) and sarcoplasmatic reticulum Ca2+ release in catecholamine-induced cardiac hypertrophy. Cardiovasc Res 2004, 61, 278-287. Li, X.; Jiang, W: Electrical remodelling of membrane ionic channels of hypertrophied ventricular myocytes from spontaneously hypertensive rats. Chin Med J 2000, 113, 584-587. Ahmmed, G.U.; Dong, P.H.; Song, G.; Ball, N.A.; Xu, Y.; Walsh, R.A.; Chiamvimonvat, N. Changes in Ca2+ cycling proteins underlie cardiac action potential prolongation in apressure-overload guinea pig model with cardiac hypertrophy and failure. Circ Res 2000, 86, 558-570. Ming, Z.; Nordin, C.; Siri, F.; Aronson, R.S. Reduced calcium current density in single myocytes isolated from hypertrophied failing guinea pig hearts. J Mol Cell Cardiol 1994, 26, 1133-1143. Ouadid, H.; Albat, B.; Nargeot, J. Calcium currents in diseased human cardiac cells. J Cardiovasc Pharmacol 1995, 25, 282-291. Wang, Z.; Kutschke, W.; Richardson, K.E.; Karimi, M.; Hill, J.A. Electrical remodeling in pressure-overload cardiac hypertrophy. Role of calcineurin. Circulation 2001, 104, 1657-1663. Ryder, K.O.; Bryant, S.M.; Hart, G. Membrane current changes in left ventricular myocytes isolated from guinea pig after abdominal aortic coarctation. Cardiovasc Res 1993, 27, 1278-1287. Brooksby, P.; Levi, A.J.; Jones, J.V. The electrophysiological characteristics of hypertrophied ventricular myocytes from spontaneously hypertensive rat. J Hypertens 1993, 11, 611-622. Cerbai, E.; Barbieri, M.; Li, Q.; Mugelli, A. Ionic basis of action potential prolongation of hypertrophied cardiac myocytes isolated from hypertensive rats of different ages. Cardiovasc Res 1994. 28, 1180-1187. Meszaros, J.; Khananshvili, D.; Hart, G. Mechanisms underlying delayed afterdepolarizations in hypertrophied left ventricular myocytes of rats. Am J Physiol Heart Circ Physiol 2001, 281, H903-914. Hill, J.A. Electrical remodeling in cardiac hypertrophy. Trends Cardiovasc Med 2003, 13, 316-322. Nattel, S.; Li, D. Ionic remodeling in the heart. Pathophysiological significance and new therapeutic opportunities for atrial fibrillation. Circ Res 2000, 87, 440-447. Shaw, R.M.; Rudy, Y. Ionic mechanisms of propagation in cardiac tissue. Roles of sodium and L-type calcium currents during reduced excitability and decreased gap junction coupling. Circ Res 1997, 81, 727-741. Sipido, K.R.; Volders, P.G.A.; Vos, M.A.; Verdonck, F. Altered Na/Ca exchange activity in cardiac hypertrophy and heart failure: a new target for therapy? Cardiovasc Res 2002, 53, 782-805. Hicks, M.N.; McIntosh, M.A.; Kane, K.A.; Rankin, A.C.; Cobbe, S.M. The electrophysiology of rabbit hearts with left ventricular hypertrophy under normal and ischemic conditions. Cardiovasc Res 1995, 30, 181-186. Winterton, S.J.; Turner, M.A.; O’Gorman, D.J.; Flores, N.A.; Sheridan, D.J. Hypertrophy causes delayed conduction in human and guinea pig myocardium: accentuation during ischemic perfusion. Cardiovasc Res 1994, 28, 47-54. Yokoshiki, H.; Kohya, T.; Tomita, F.; Tohse, N.; Nakaya, H.; Kanno, M.; Kitabatake, A. Restoration of action potential duration and transient outward current by regression of left ventricular hypertrophy. J Mol Cell Cardiol 1997, 29, 1331-1339. Aiello, E.A.; Villa-Abrille, M.C.; Escudero, E.M.; Portiansky, E.L.; Perez, N.G.; de Hurtado, M.C.C.; Cingolami, H.E. Myocardial hypertrophy of normotensive Wistar-Kyoto rats. Am J Physiol Heart Circ Physiol 2004, 286, H1229-H1235. Cooklin, M.; Wallis, W.R.J.; Sheridan, D.J.; Fry, C.H. Changes in cell-to-cell electrical coupling associated with left ventricular hypertrophy. Circ Res 1997, 80, 765-771. Wiegerinck, R.F.; Verkerk, A.O.; Belterman, C.N.; van Veen, T.A.; Baartscheer, A.; Opthof, T.; Wilders, R.; de Bakker, J.M.; Coronel, R. Larger cell size in rabbits with heart failure increases myocardial conduction velocity and QRS duration. Circulation 2006, 113, 806-813.

Minor comment

Figure 2: I couldn't distinguish between thick and thin arrows. Please make the difference between the two easy to understand.

Figure 2: The thickness of the “thick” arrows was increased to make the difference more visible.

Reviewer 2 Report

Dear author !

thanks for the nice paper. It seems you have enough expertise in the topic, taking into account your recent publications. In this "problem" review you tried to rise the link between basic science morphological knowledge about LV remodeling in LVH and the absence of clear functional electrphysiological characteristics of the electrical studies. From this point of view the study is very actual.

From one side you present morphological and microelectrode studies of the hypertrophed myocardium and from another point of view: QRS complex duration, QT-interval,early repolirization, LV potentials, Fragmente QRS complex, classic ECG, etc. At the end there is the conclusion about missing link between 2 spheres of knowlegde.

The paper is well edited, have a number of current citations, and at the end the question for further consideration is reasonable.

However I would like to add:

1) As you described some electrphisiological methods, you can increase their number by for example body surface potential mapping, Heart Rate Turbulence from one side and patch clamp (change of ion currents) from another.

2) I would make more detailed Figure, which integrate the existing knowledge that you described and methods of investigation and show that is missing in details. Now "Conclusions" looks very general.

I would like to see the paper after the editing.

Author Response

Reviewer 2:

Dear author !

thanks for the nice paper. It seems you have enough expertise in the topic, taking into account your recent publications. In this "problem" review you tried to rise the link between basic science morphological knowledge about LV remodeling in LVH and the absence of clear functional electrophysiological characteristics of the electrical studies. From this point of view the study is very actual.

From one side you present morphological and microelectrode studies of the hypertrophied myocardium and from another point of view: QRS complex duration, QT-interval, early repolarization, LV potentials, Fragmented QRS complex, classic ECG, etc. At the end there is the conclusion about missing link between 2 spheres of knowledge.

The paper is well edited, have a number of current citations, and at the end the question for further consideration is reasonable.

However I would like to add:

As you described some electrophysiological methods, you can increase their number by for example body surface potential mapping, Heart Rate Turbulence from one side and patch clamp (change of ion currents) from another.

Thank you for suggesting me to include more detailed information on ECG methods on the one hand and ion channels on the other hand. When writing the manuscript I had a dilemma how much details to present on the ion channels remodeling in LVH, and I was afraid that going into details the review would be not balanced. On the other hand I understand that it is the key topic. The knowledge on ion channels is really extensive and I did my best to compress it into a limited space to balance it with the other parts of the manuscript. I am sure other experts will provide more qualified details on this issue.

The following two paragraphs on ion channels were added with corresponding references [34-53]:

Page 3, lines 100-115:

Animal as well as human studies show a variety of results on the main depolarizing currents affecting the resting potential and the action potential configuration : the fast sodium current INa , L-type calcium current ICa(L) and Na+-Ca2+ exchanger INa/Ca. A decrease in INa density was documented [34, 35], on the other hand its increase was described [36]. Similarly, a decrease in ICa(L current was shown [37, 38], an increase [39, 40], or no difference was reported [41, 42]. More consistent are findings regarding INa/Ca, where an increase in I(Na/Ca) density was reported [34, 43]. The ion channel remodeling are reviewed in details by e.g. [44-47].

Alterations in ion channels are reflected in changes in resting membrane potential, as well as in action potential (AP). The results related to the depolarization phase of AP are not consistent. On one hand, no differences in resting membrane potential, the action potential upstroke velocity and amplitude are documented [48, 49], on the other hand, significant changes are reported [40, 50, 51]. Also, there is not a consistent pattern in conduction velocity changes of hypertrophied cardiomyocytes [52, 53]. The variety of results can be due to differences in experimental models of LVH, species and preparation employed, the severity of hypertrophy, methodological aspects, etc., however, they create a solid base of evidence on the changes influencing electrical characteristics of hypertrophied cardiomyocytes and myocardium.

Chorvatova, A.; Hart, G.; Hussain, M. Na+/Ca2+ exchange current (I(Na/Ca)) and sarcoplasmatic reticulum Ca2+ release in catecholamine-induced cardiac hypertrophy. Cardiovasc Res 2004, 61, 278-287. Li, X.; Jiang, W: Electrical remodelling of membrane ionic channels of hypertrophied ventricular myocytes from spontaneously hypertensive rats. Chin Med J 2000, 113, 584-587. Ahmmed, G.U.; Dong, P.H.; Song, G.; Ball, N.A.; Xu, Y.; Walsh, R.A.; Chiamvimonvat, N. Changes in Ca2+ cycling proteins underlie cardiac action potential prolongation in apressure-overload guinea pig model with cardiac hypertrophy and failure. Circ Res 2000, 86, 558-570. Ming, Z.; Nordin, C.; Siri, F.; Aronson, R.S. Reduced calcium current density in single myocytes isolated from hypertrophied failing guinea pig hearts. J Mol Cell Cardiol 1994, 26, 1133-1143. Ouadid, H.; Albat, B.; Nargeot, J. Calcium currents in diseased human cardiac cells. J Cardiovasc Pharmacol 1995, 25, 282-291. Wang, Z.; Kutschke, W.; Richardson, K.E.; Karimi, M.; Hill, J.A. Electrical remodeling in pressure-overload cardiac hypertrophy. Role of calcineurin. Circulation 2001, 104, 1657-1663. Ryder, K.O.; Bryant, S.M.; Hart, G. Membrane current changes in left ventricular myocytes isolated from guinea pig after abdominal aortic coarctation. Cardiovasc Res 1993, 27, 1278-1287. Brooksby, P.; Levi, A.J.; Jones, J.V. The electrophysiological characteristics of hypertrophied ventricular myocytes from spontaneously hypertensive rat. J Hypertens 1993, 11, 611-622. Cerbai, E.; Barbieri, M.; Li, Q.; Mugelli, A. Ionic basis of action potential prolongation of hypertrophied cardiac myocytes isolated from hypertensive rats of different ages. Cardiovasc Res 1994. 28, 1180-1187. Meszaros, J.; Khananshvili, D.; Hart, G. Mechanisms underlying delayed afterdepolarizations in hypertrophied left ventricular myocytes of rats. Am J Physiol Heart Circ Physiol 2001, 281, H903-914. Hill, J.A. Electrical remodeling in cardiac hypertrophy. Trends Cardiovasc Med 2003, 13, 316-322. Nattel, S.; Li, D. Ionic remodeling in the heart. Pathophysiological significance and new therapeutic opportunities for atrial fibrillation. Circ Res 2000, 87, 440-447. Shaw, R.M.; Rudy, Y. Ionic mechanisms of propagation in cardiac tissue. Roles of sodium and L-type calcium currents during reduced excitability and decreased gap junction coupling. Circ Res 1997, 81, 727-741. Sipido, K.R.; Volders, P.G.A.; Vos, M.A.; Verdonck, F. Altered Na/Ca exchange activity in cardiac hypertrophy and heart failure: a new target for therapy? Cardiovasc Res 2002, 53, 782-805. Hicks, M.N.; McIntosh, M.A.; Kane, K.A.; Rankin, A.C.; Cobbe, S.M. The electrophysiology of rabbit hearts with left ventricular hypertrophy under normal and ischemic conditions. Cardiovasc Res 1995, 30, 181-186. Winterton, S.J.; Turner, M.A.; O’Gorman, D.J.; Flores, N.A.; Sheridan, D.J. Hypertrophy causes delayed conduction in human and guinea pig myocardium: accentuation during ischemic perfusion. Cardiovasc Res 1994, 28, 47-54. Yokoshiki, H.; Kohya, T.; Tomita, F.; Tohse, N.; Nakaya, H.; Kanno, M.; Kitabatake, A. Restoration of action potential duration and transient outward current by regression of left ventricular hypertrophy. J Mol Cell Cardiol 1997, 29, 1331-1339. Aiello, E.A.; Villa-Abrille, M.C.; Escudero, E.M.; Portiansky, E.L.; Perez, N.G.; de Hurtado, M.C.C.; Cingolami, H.E. Myocardial hypertrophy of normotensive Wistar-Kyoto rats. Am J Physiol Heart Circ Physiol 2004, 286, H1229-H1235. Cooklin, M.; Wallis, W.R.J.; Sheridan, D.J.; Fry, C.H. Changes in cell-to-cell electrical coupling associated with left ventricular hypertrophy. Circ Res 1997, 80, 765-771. Wiegerinck, R.F.; Verkerk, A.O.; Belterman, C.N.; van Veen, T.A.; Baartscheer, A.; Opthof, T.; Wilders, R.; de Bakker, J.M.; Coronel, R. Larger cell size in rabbits with heart failure increases myocardial conduction velocity and QRS duration. Circulation 2006, 113, 806-813.

Regarding the ECG methods, the review is focused on the evaluation of the QRS complex using the standard 12-lead ECG. Since the body surface potential mapping is very promising method with a great potential, I was happy to include the following paragraph with corresponding references:

Page 7, lines 244-253:

Body surface potential mapping (BSPM) provides total information on the temporo-spatial distribution of electrical potentials on the surface of the chest. As compared to limited information provided by the standard 12-lead ECG, it enables detection of local electrical events or local conduction disturbances, reflecting heterogeneity of underlying ventricular activation and repolarization [118]. Thus it can identify the signs of susceptibility to arrhythmias and the site of origin of the arrhythmia. Additional advantage is a possibility to derived epicardial maps non-invasively (ECG-imaging) from the distribution of surface potentials [119].

Although BSPM is a promising method, its utilization in clinical practice is still limited because of its demanding recording and interpretation, as well as of the need for standardization of lead systems and methods of BSPM data analysis.

De Ambroggi, L.; Corlan, A.D. Clinical use of body surface potential mapping in cardiac arrhythmias. Anatol J Cardiol 2007, 7 Suppl 1, 8-10. Burnes, J.E.; Taccardi, B.; Rudy, Y. A noninvasive imaging modality for cardiac arrhythmias. Circulation 2000, 102, 2152–2158.

If you agree, I would like to keep the review focused on the QRS complex measures, and will not include the heart rate turbulence which would open another extensive topic on the heart rate evaluation, heart rate variability, and ventricular ectopic beats.

I would make more detailed Figure, which integrate the existing knowledge that you described and methods of investigation and show that is missing in details. Now "Conclusions" looks very general.

The Figure 4 was included summarizing the described methods and hopefully showing better the relations in LVH.

Figure 4. The main factors influencing electrogenesis in left ventricular hypertrophy and the variety of ECG manifestations related to ventricular depolarization (QRS complex). fQRS: fragmented QRS complex, LP: late ventricular potential, ER: early repolarization, LAD: left axis deviation, LAFB: left anterior fascicular block, LBBB: left bundle branch block.

Round 2

Reviewer 1 Report

Thank you for responding to my suggestions.

Figure 2: It became easier to distinguish between thick and thin arrows. Please correct the thick arrows in Figure 2C as well.

Author Response

Dear Reviewer:

Thank you for your careful evaluation. I have corrected the Figure 2C, and I am sorry for my mistake.
